# Prevalence of Carbendazin Resistance in Field Populations of the Rice False Smut Pathogen *Ustilaginoidea virens* from Jiangsu, China, Molecular Mechanisms, and Fitness Stability

**DOI:** 10.3390/jof8121311

**Published:** 2022-12-16

**Authors:** Jiehui Song, Zhiying Wang, Yan Wang, Sijie Zhang, Tengyu Lei, You Liang, Qigen Dai, Zhongyang Huo, Ke Xu, Shuning Chen

**Affiliations:** 1Jiangsu Key Laboratory of Crop Genetics and Physiology & Co-Innovation Center for Modern Production Technology of Grain Crops, Agricultural College, Yangzhou University, Yangzhou 225009, China; 2Key Laboratory of Pesticides Evaluation, Ministry of Agriculture, Institute of Plant Protection, Chinese Academy of Agricultural Sciences, Beijing 100193, China

**Keywords:** β-tubulin, carbendazim resistance, fitness, molecular mechanism, *Ustilaginoidea virens*

## Abstract

Rice false smut (RFS), caused by *Ustilaginoidea virens*, is an important fungal disease of rice. In China, Methyl Benzimidazole Carbamates (MBCs), including carbendazim, are common fungicides used to control RFS and other rice diseases. In this study, resistance of *U. virens* to carbendazim was monitored for three consecutive years during 2018 to 2020. A total of 321 *U. virens* isolates collected from Jiangsu Province of China were tested for their sensitivity to carbendazim on PSA. The concentration at which mycelial growth is inhibited by 50% (EC_50_) of the carbendazim-sensitive isolates was 0.11 to 1.38 µg/mL, with a mean EC_50_ value of 0.66 μg/mL. High level of resistance to carbendazim was detected in 14 out of 321 isolates. The resistance was stable but associated with a fitness penalty. There was a statistically significant and moderate negative correlation (r= −0.74, *p* < 0.001) in sensitivity between carbendazim and diethofencarb. Analysis of the *U. virens* genome revealed two potential MBC targets, *Uvβ1Tub* and *Uvβ2Tub*, that putatively encode β-tubulin gene. The two β-tubulin genes in *U. virens* share 78% amino acid sequence identity, but their function in MBC sensitivity has been unclear. Both genes were identified and sequenced from *U. virens* sensitive and resistant isolates. It is known that mutations in the β2-tubulin gene have been shown to confer resistance to carbendazim in other fungi. However, no mutation was found in the *Uvβ2Tub* gene in either resistant or sensitive isolates. Variations including point mutations, non-sense mutations, codon mutations, and frameshift mutations were found in the *Uvβ1Tub* gene from the 14 carbendazim-resistant isolates, which have not been reported in other fungi before. Thus, these results indicated that variations of *Uvβ1Tub* result in the resistance to carbendazim in field isolates of *Ustilaginoidea virens*.

## 1. Introduction

Rice false smut (RFS), caused by the filamentous fungus *Ustilaginoidea virens* (teleomorph: *Villosiclava virens*), is responsible for significant losses in the rice industry. RFS has long been a minor disease in rice production and occurs sporadically in some rice-growing regions, such as south and east Asia. However, it has recently become one of the most important diseases in most rice-growing regions of the world [1]. The false smut pathogen (*U. virens*) infects the plant during the flowering stage [2]. The infected grains convert first into whitish, yellowish-orange to green chlamydospores, which later turn greenish-black in colour [3]. Worse still, the *U. virens* can produce cyclopeptide mycotoxins, which are poisonous to humans and livestock and pose a serious problem for food security and rice production [4,5].

To date, although some RFS-resistant rice cultivars are commercially produced, disease control of RFS relies mainly on fungicide application [1]. An ideal prevention effect can be achieved when fungicides are applied one week before heading, which is an important infection stage for RFS [6,7,8]. The fungicides extensively used for RFS control in China include the Methyl Benzimidazole Carbamates fungicides (carbendazim and thiophanate-methyl), sterol demethylation inhibitors (DMIs) fungicides (difenoconazole, prochloraz, propiconazole, simeconazole, and tebuconazole), the quinone outside inhibitor (QoI) fungicides (azoxystrobin and pyraclostrobin), copper oxychloride, and Jinggangmycin. 

Of the commonly used site-specific fungicides, the status of the fungicide resistant and control efficacy of DMIs and QoI towards *U. virens* have been investigated. As for the DMI fungicides, laboratory tebuconazole-resistant *U. virens* mutants were firstly reported by both UV-mutagenesis and fungicide-taming methods [9,10]. Then, resistance was also found in the field, where 2 of 158 tested field isolates highly resistant to propiconazole were reported [11]. In terms of the QoI fungicides, the baseline sensitivities of *U. virens* towards azoxystrobin and pyraclostrobin have been established [12]. Although QoI fungicides were categorized as high risk for resistance established by the Fungicide Resistance Action Committee [13], the application of QoI fungicides to control RFS is at low risk at present [12].

MBC fungicides were introduced into the market in the early 1970s and became widely used in agriculture [14]. The fungicides exert their antifungal activity by targeting the β-tubulin subunit of the microtubules. MBC fungicides currently available include benomyl, carbendazim, thiabendazole, and thiophanate-methyl. Among them, carbendazim combined with tebuconazole, hexaconazole, and triadimefon has been registered for the control of RFS in China and has been extensively used since the outbreak of the RFS. In addition to the control of RFS, MBC fungicides were also frequently used for the control of other rice diseases, such as rice blast, sheath blight of rice, and bakanae disease of rice. According to the registration information, carbendazim is the second-most-used fungicide on rice (China Pesticide Information Network, http://www.chinapesticide.org.cn accessed on 16 December 2022). Farmers may apply 3–4 sprays of MBC fungicides on the rice per season for the control of rice diseases. Unfortunately, MBC fungicides, as typical site-specific fungicides, are prone to resistance development. Resistance to benzimidazole fungicides has been reported in *Botrytis cinerea*, *Fusarium graminearum*, *Helminthosporium solani*, and many other plant-pathogenic fungi, and usually results from mutations at Phe167Tyr, Glu198Ala, Val, Gly, Phe200Tyr, and other codons in the β-tubulin gene (*Tub*) [15,16,17,18]. In addition, carbendazim may cause endocrine disruption in wildlife and cancer in humans [19]. Carbendazim has been banned in some countries, such as the USA and the UK. Therefore, the application of carbendazim in agriculture might be restricted or prohibited in China in the future.

The sensitivity of *U. virens* to MBC fungicides in China is unclear, and carbendazim-resistant mutants have not been characterized. Therefore, the objectives of this paper were: (1) determine the carbendazim sensitivity of *U. virens* isolates from different rice field areas around the major rice-producing area, Jiangsu Province; (2) characterize the carbendazim-resistant isolates; and (3) understand the molecular mechanisms of the carbendazim-resistant isolates.

## 2. Materials and Methods

### 2.1. Isolation

Isolates of *U. virens* used in this study were obtained during 2018 to 2020 from Jiangsu Province, which is the major rice production province in China. The rice fields are located in 10 counties and cities, namely Gaoyou, Jurong, Nanjing, Taizhou, Xinhua, Xuyi, Yizheng, Yangzhou Guanglin (YZ Guanglin), Yangzhou Hanjiang (YZ Hanjiang), and Zhenjiang (Figure 1). Rice varieties grown in those areas include Wanxian No.98, Najing No. 9108, Nanjing No. 46, Nanjing No. 5718, Nanjing No. 5055, Yangchannuo No. 1, Huiliangyou No. 898, and Yongyou No. 2640. Three to five rice plants with symptoms of rice false smut were randomly selected from various blocks in each field and preserved at 4 °C. Individual false smut balls collected from different rice plants served as the source for all isolates. The false smut balls were divided in half and placed on PSA plates amended with streptomycin sulfate after being surface-sterilized in 0.1% sodium hypochlorite for 5 min. The plates were cultured in a growth chamber for 3–7 days at 27 °C (12 h photoperiod). Pure cultures were obtained by transfer of a single germinated chlamydospore and maintained on PSA slants at 4 °C for 2–4 weeks. A total of 321 isolates were collected throughout Jiangsu Province. 

### 2.2. Fungicides and Medium 

Technical-grade carbendazim (97% active ingredient [a.i.]; Jinbei Chemical Co., Ltd., Jingzhou, Hubei, China), azoxystrobin (96% a.i.; Jiangsu Gengyun Chemical Co., Ltd., Changzhou, Jiangsu, China), pyraclostrobin (95% a.i.; Hubei Kang Bao Tai Fine-Chemical Co. Ltd., Wuhan, Hubei, China), tebuconazole (96% a.i.; Sheyang Huanghai Pesticides and Chemical Co., Ltd., Yancheng, Jiangsu, China), and diethofencarb (95.4% a.i.; Jiangsu Lanfeng Chemical Co., Ltd., Xuzhou, Jiangsu, China) were used in this study. Stock solutions were formulated by dissolving each fungicide in methanol at the concentration of 10,000 µg a.i./mL, except for carbendazim, which were dissolved in 0.2 mol/L HCl. PSB was prepared with 200 g of potato and 20 g of sugar per liter of water. PSA was prepared by adding 15 g of agar per liter of PSB.

### 2.3. In Vitro Sensitivity Determination of U. virens to Carbendazim 

Sensitivity to carbendazim was assessed on fungicide-amended PSA at 0, 0.03, 0.1, 0.3, 1, 3, 10, 30, and 60 µg a.i./mL. Carbendazim-resistant isolates were subjected to a second series of sensitivity assays involving PSA amended with carbendazim at 0, 100, 250, and 500 µg a.i./mL. To inoculate test plates, mycelial plugs were removed with a 4 mm cork borer from the margins of 14-day-old colonies and placed upside down on the centers of 9 cm plastic Petri dishes containing the fungicide-amended or unamended media. Each isolate was tested three times, and the plates were incubated at 27 °C for 21 days in darkness. For each treatment, the mean colony diameter (minus the diameter of the inoculation plug) was measured and expressed as a percentage of growth inhibition. The EC_50_ values were calculated by regression analysis of probit values of corresponding percent growth inhibitions against logarithms of fungicide concentrations. Baseline sensitivity of carbendazim was constructed on the basis of frequency distribution of logarithmically transformed EC_50_ values of the 321 isolates.

### 2.4. Investigation of Mycelial Growth, Mycelial Dry Weight, Conidiation, and Conidial Germination Rate In Vitro

Mycelial growth diameters of 14 resistant isolates and 5 sensitive isolates were determined on fungicides-free PSA plates. Mycelium plugs (4 mm in diameter) taken from the periphery of 14-day-old cultures were transferred to new PSA plates and incubated at 27 °C in the dark. The colony diameters were measured after 14 days, with three replicates per isolates. Fourteen-day-old mean colony diameters of all resistant isolates or sensitive isolates were calculated as mycelial growth diameters. For conidiation test and measurement of conidia germination, after isolates were cultured on PSA at 27 °C for 14 days, two 4 mm diameter mycelia plugs were transferred into 50 mL of PSB, and the conidiation was counted with a hemocytometer after shaking at 27 °C, 150 rpm for 7 days. The mycelial were collected by filtration through two layers of gauze and measured for dry weight after drying at 50 °C in an oven for 3 days. To measure conidial germination rate, mycelial were removed by filtration and the conidia were collected from the filtrate by centrifugation at 7000× *g* for 5 min. The conidia were washed twice by resuspension in sterile distilled water, in which they were finally resuspended and adjusted to a concentration of 1.0 × 10^5^ conidia/mL. Then, 100 μL conidia suspension was spread on the surface of PSA or WA (water agar: 15 g agar per liter water) plates, and the plates were incubated at 27 °C in the dark for 24 h. The conidial germination rate was recorded with a Nikon E200 microscope. Conidia was regarded as germinated when the germ tube length exceeded half the conidia length. Germination was quantified by counting at least 200 conidia per plate for each isolate. Each isolate was repeated three times and each experiment was conducted twice.

### 2.5. Stability of Resistant to Carbendazim

Mycelium plugs were taken from the periphery of the colonies and transferred to fresh fungicide-free PSA, then the plate was incubated at 27 °C for 10 days. The process was repeated for 5 generations, but new plates were always inoculated with the colony of the previous generation. EC_50_ values for carbendazim were established before the first and after the last transfer, as described above. 

### 2.6. Correlation Analysis in Sensitivity of Carbendazim with Diethofencarb, Azoxystrobin, Pyraclostrobin, and Tebuconazole

For the correlation analysis assay, the sensitivity of nine carbendazim-resistant mutants and nine carbendazim-sensitive isolates to diethofencarb, azoxystrobin, pyraclostrobin, and tebuconazole were determined by measuring mycelial growth. Sensitivity to diethofencarb, azoxystrobin, pyraclostrobin, and tebuconazole was assessed on fungicide-amended PSA at 0, 0.03, 0.1, 0.3, 1, 3, 10, and 30 µg a.i./mL. For carbendazim-sensitive isolates, PSA media were amended with diethofencarb at final concentrations of 0, 10, 30, 100, 250, and 500 µg a.i./mL. In sensitivity determination to azoxystrobin and pyraclostrobin, salicylhydroxamic acid (SHAM) was not included in the medium, since all tested isolates were sensitive to these two QoI fungicides, and SHAM showed strong toxicity to *U. virens*, according to our previous studies [12,20]. The EC_50_ value was estimated as previously described. The experiments were performed three times with four replicates plates per concentration. The cross-resistance relationships were analyzed by the correlation procedure in GraphPad Prism (version 5.01, GraphPad Software, San Diego, CA, USA).

### 2.7. Cloning and Sequence Analysis

Fourteen carbendazim resistant isolates and twenty-three sensitive isolates collected from different cities were selected for *Uvβ1Tub* and *Uvβ2Tub* analysis. All of them were grown on PSA at 27 °C for 14 days in the dark. Single agar plugs containing actively growing mycelium were transferred to 100 mL flasks containing 50 mL of PSB. Flasks were shaken at 150 rpm for 7 days at 27 °C. Mycelium was filtered from the broth, rinsed under sterile deionized water, and subjected to DNA extraction using the DNeasy Plant Mini Kit (Qiagen Inc., Valencia, CA, USA) according to the manufacturer’s instructions. 

On the basis of the whole nucleotide sequence of *Uvβ1Tub* (Gene ID: UV8b_05680) and *Uvβ2Tub* (Gene ID: UV8b_05383) from the sequenced isolate UV8b, the primer pairs tub1-5N/tub1-3N: ACAGTGATGCGTGATGCGAT/TGTTGGCTCAACGAGGTCAA were designed to amplify the *Uvβ1Tub* and its 711-bp upstream and 459-bp downstream fragment, and the primer pairs tub2-F/tub2-R: GGTACTCCGTAAACGTAATC/TCACCCTTCTGCTGGTTGCG were designed to amplify the *Uvβ2Tub* and its upstream fragment from the isolates analyzed. The polymerase chain reaction (PCR) mixtures contained 1 × PCR buffer, 20 ng of template DNA, 0.6 mM each primer, 200 mM each dNTP, and 1 U of Taq DNA polymerase (New England Biolabs, Ipswich, MA, USA). The following PCR parameters were used: an initial denaturation step of 5 min at 94 °C; followed by 35 cycles of 30 s at 94 °C, 30 s at 55 °C, and 3 min at 72 °C; and a final extension step of 7 min at 72 °C. 

### 2.8. Phylogenetic Analysis

Amino acid sequences of β1Tub and β2Tub from *U. virens* and other plant pathogens were obtained from the public database GenBank. Multiple alignments were conducted using DNASTAR (DNASTAR Inc., Nevada City, CA, USA) and CLUSTAL X v. 2.1. For the ML method, the phylogram was inferred on the basis of the JTT matrix-based model. The bootstrap consensus tree, inferred from 1000 replicates, was constructed with the following parameters: the Poisson correction model, gamma distribution (five categories), and heuristic method using SPR-extensive. All positions containing gaps and missing data were eliminated.

### 2.9. Data Analysis

Significant differences of EC_50_ values from different populations were evaluated by one-way analysis of variance with a least significant difference test in SPSS Software (version 22.0; IBM SPSS Inc. Chicago, IL, USA). To determine cross-resistance in isolates to fungicides, the EC_50_ values were correlated, and the correlation coefficients (r) were calculated by SPSS. There were three replicates of each concentration for each isolate. 

## 3. Results

### 3.1. Sensitivity of the Field Isolates to Carbendazim

In total, 321 isolates were collected from 10 cities in Jiangsu Province. The average frequency of resistance to carbendazim was 4.36% (Table 1). Except for Gaoyou, Nanjing, and Taizhou, carbendazim-resistant isolates were found in all of the other cities. The EC_50_ values of the resistant isolates were more than 500 μg/mL for most of the resistant isolates. The resistance factors (defined as the ratio of EC_50_ for a fungicide resistant isolate relative to the mean EC_50_ of the sensitive isolates) were approximately 100 or even higher. 

However, as for the sensitive isolates, carbendazim was quite effective, with EC_50_ range from 0.108 to 1.378 µg/mL and mean EC_50_ value of 0.663 μg/mL (Table 1). Among the sensitive isolates, there was no difference (*p* > 0.05) in mean EC_50_ values of isolates from different cities. 

The frequency distribution of the EC_50_ values for carbendazim was unimodal (Figure 2A). In terms of years, although no resistant isolates were detected in 2020, the proportion of isolates insensitive to carbendazim in 2020 was higher than those in 2018 and 2019 (Figure 2B).

### 3.2. Fitness and Stability of Resistant Isolates

Four fitness components were tested for all 14 resistant isolates and 5 sensitive isolates. Overall, fitness penalties were recorded for the resistant isolates. The mycelial growth diameters of the resistant isolates were significantly lower than those of the sensitive isolates (Figure 3A). In addition, mycelial dry weight was lower in the resistant isolates, but the difference was statistically non-significant (Figure 3B, *p* > 0.05). The resistant isolates produced significantly fewer conidia in PSB compared with the sensitive isolates (Figure 3C). Since most of resistant isolates produced no or little conidia, five resistant isolates (GL11, HA17, XH5a, XH7b, and XH43b) were tested for conidial germination rate on PSA and WA media. The conidial germination rate of the resistant isolates was also significantly lower than that of the sensitive isolates (Figure 3D). 

The stability of resistance to carbendazim in 14 isolates was determined by culturing the isolates on fungicide-free PSA medium. Unchanged resistance factors before (T0) and after five generations (T5) indicated that resistance was stable (Figure 4 and Table 2). 

### 3.3. Correlation in Sensitivity of Carbendazim to Diethofencarb, Azoxystrobin, Pyraclostrobin, and Tebuconazole

As shown in Figure 5A, based on the EC_50_ values, the correlation coefficient between carbendazim–diethofencarb was −0.74 (*p* < 0.001), indicating a moderate but significant negative cross-resistance between them. As for other fungicides used to control rice false smut, none of the carbendazim-resistant mutants exhibited resistance to azoxystrobin, pyraclostronbin, or tebuconazole. Spearman rank correlations in sensitivity between carbendazim and those three fungicides were analyzed, and no significant correlation was detected in sensitivity between carbendazim and azoxystrobin (r = 0.24, *p* = 0.33), pyraclostronbin (r = 0.29, *p* = 0.24), or tebuconazole (r = −0.32, *p* = 0.19) (Figure 5B–D).

### 3.4. Nucleotide Sequence Analysis of the β-Tubulin Genes

Both *Uvβ1Tub* and *Uvβ2Tub* genes and their upstream fragments in the resistant and sensitive isolates were amplified and sequenced. Twenty-three sensitive isolates collected from different cities have the same amino acid sequences in both genes. The nucleotide sequences of *Uvβ1Tub* amplified from the twenty-three sensitive isolates, including exons and introns, were 1825 bp in length. The full-length cDNA of *Uvβ1Tub* amplified from RNA of the sensitive isolates was 1347 bp in length, encoding a putative polypeptide of 448 amino acids. Sequence comparison of the genomic DNA and cDNA revealed four introns located at nucleotide position 13–241, 266–333, 457–571, and 1363–1428. The *Uvβ2Tub* from carbendazim-sensitive isolates nucleotide sequences (including exon and intron) was 2103 bp. The full-length cDNA of *Uvβ2Tub* was 1359 bp in length, encoding a putative polypeptide of 452 amino acids. Sequence comparison of the genomic DNA and cDNA revealed six introns located at nucleotide position 13–156, 181–259, 328–492, 548–646, 960–1079, and 1379–1515 (Figure 6A). 

The two β-tubulin genes in *U. virens* exhibit high sequence identity (78.12%), especially at the MBC-binding sites (amino acid residues 6, 165 to 167, 198 to 200, 240, and 241) (Figure 6B).

There was no amino acid difference between the carbendazim-sensitive isolates and carbendazim-resistant in the Uvβ2Tub. As for the Uvβ1Tub, compared with the sensitive isolates, several variations were found in the resistant isolates. The variations included point mutations, non-sense mutations, codon mutations, and frameshift mutations (Table 3). As for point mutations, isolate GL11 had a variation at codon 91, resulting in an amino acid alteration from V to G, and isolate ZJ7 had a variation from A to D at codon 411. Two isolates, GL12b and JR12, had non-sense mutations. For GL12b, the mutation is a C-to-T transversion at nucleotide 871, transforming the codon 291 from Glutamine to a stop codon. For JR12, the mutation at nucleotide 63 (G to A) resulted in a stop codon at amino acid 21. Two isolates had codon mutations, JR11 had 15-bp deletion at codon 408–412, and YD8 had 3-bp deletion at codon 221. Five isolates had frameshift mutations, resulting in a change to a gene’s reading frame—one to seven deletions were found in those isolates. The locations where mutations were found from different resistant isolates on the *UvTub1* are marked in Figure 4A,B. The codon positions where mutations have been reported from other species in *β1Tub* and *β2Tub* are marked in Figure 4B. All of the mutations found in our study have never before been reported from other pathogens.

In addition, for isolates HA26, XH7b, and ZJ24, no mutations were found in *Uvβ1Tub* and *Uvβ2Tub*, nor in their upstream fragments. All of the sequences amplified in this study have been deposited in Genbank (Appendix A).

### 3.5. Phylogenetic Analyses of Predicted Amino Acid Sequences

The deduced amino acid sequence of Uvβ1Tub was 99.07% identical to that of *Metarhizium humberi* (KAH0601712), 99.07% identical to that of *Epichloe typhina* (P17938), 98.61% to that of *Fusarium albosuccineum* (KAF4455974), and 97.91% to that of *Clonostachys rosea* (VUC22392). Comparative analysis of the sequences yielded an E-value of 0.0, which confirmed Uvβ1Tub to be a member of the fungal β1Tub family.

Comparison of the Uvβ2Tub with those of other filamentous fungus indicated that it shared the greatest identity with the sequences of *Epichloe gansuensis* (93.94%) (AJQ24524.1), *Claviceps maximensis* (92.56%) (KAG6002942.1), *Fusarium solani* (83.49%) (UNA06309.1), and *Botryosphaeria dothidea* (82.06%) (QWT72326.1). The percentage identity confirmed Uvβ2Tub to be a member of the filamentous fungus β2Tub family.

A phylogenetic tree was constructed on the basis of the concatenated alignment of β-tubulin homologs of *U. virens* and other ascomycete fungi that are close to *U. virens*, including the outgroup fungus *Candida glabrata* (Figure 7). There were a total of 443 positions in the final data set. Maximum likelihood phylogenetic analyses of the predicted amino acid sequences for these two proteins (β1Tub and β2Tub) with sequences for 36 *Tub* genes from ascomycete species resolved two distinct clades, with β1Tub and β2Tub forming two monophyletic clades. Results confirmed that Uvβ1Tub and Uvβ2Tub were homologous to the β1Tub and β2Tub proteins from multiple other fungi, respectively.

## 4. Discussion

Fungicide application has been an effective option for controlling rice diseases in China. MBC fungicides, such as carbendazim and thiophanate-methyl, which were extensively used for the control of rice blast, sheath blight of rice, and bakanae disease of rice, were recently adapted for controlling RFS. However, few studies have reported the sensitivity of field isolates of *U. virens* to MBC fungicides. In this study, for the first time, the resistance of *U. virens* towards MBCs was monitored, and the resistance isolates within field population of *U. virens* were characterized.

A total of 321 isolates were collected from the main rice production area of China, and their sensitivities to the MBC fungicides were determined. The results showed the overall frequency of MBC resistant isolates was 4.36%, and all of the resistant isolates were of high resistance level. Except for the resistant isolates, other isolates were very sensitive to carbendazim, with a mean EC_50_ value of 0.66 μg/mL. The results were similar to those for the *Gibberella zeae* and *Fusarium* species complexes causing pokkah boeng disease, with mean EC_50_ values of 0.59 μg/mL and 0.60 μg/mL, respectively [25,26]. Although the sensitivity tested in this study could not be treated as the baseline, as the isolates have been previously exposed to MBC fungicides, the information provides a frame of reference for future issues with MBC insensitivity in *U. virens*.

The correlation analysis assay showed that the nine resistant isolates were sensitive to azoxystrobin, pyraclostrobin, and tebuconazole, indicating that there was no multiple fungicide resistance among MBC, QoI, and DMI fungicides. Thus, QoI and DMI fungicides can be used for the management of carbendazim resistance in *U. virens.* Moreover, the mixture of carbendazim with DMIs, such as tebuconazole (registration number: PD20110332), triadimefon (registration number: PD20060057), and hexaconazole (registration number: PD20181518), which has been registered to control rice diseases in China, could still be used to manage the resistance of *U. virens* to MBC at present. In addition, an important anti-resistance strategy would be to limit MBC fungicide sprays to a minimum and only in mixtures with multi-site, broad spectrum, protectant, low-risk fungicides [27].

A moderate negative cross-resistance between carbendazim and diethofencarb was observed in carbendazim-resistant isolates. Strong negative cross relationships could be observed between diethofencarb and benzimidazole-resistant isolates with E198A and E198V mutations [28,29]. In several pathogens, such as in *Botrytis* spp. and *Monilinia* spp., diethofencarb inhibits benzimidazole-resistant but not benzimidazole-sensitive isolates [30,31,32]. However, in *U. virens*, benzimidazole-sensitive isolates were also largely sensitive to diethofencarb (with EC_50_ values of approximately 10 μg/mL). The binding mode between UvTub and diethofencarb needs to be further investigated. 

Our results also indicate that the acquisition of MBC resistance is accompanied by a reduction in fitness, in that the resistance isolates grew more slowly, produced fewer conidia, and the conidia germinated less than the sensitive isolates. Fitness plays a significant role in the evolution of fungicide resistance in the fungal population. Thus, the investigation of fitness is important for the establishment of effective strategies for resistance management. In several pathogens, MBC-resistant isolates shared similar fitness with wild-type isolates and suffered little fitness penalty [32,33,34,35,36]. Similar to our results, fitness reduction was also observed in *F. fujikuroi* MBC-resistant isolates in terms of fewer conidia and less virulence [23]. In addition, MBC resistant isolates bearing different mutations might have different fitness, since different point mutation isolates possessed different predominance. Take *B. cinerea* for example; MBC resistant isolates collected from China usually harbored the E198A/V/K mutation. Among them, E198V was quite predominant compared with E198A/K [33,34,37]. In Japan, E198V and E198A were widely determined, while E198K and F200Y were barely found [29]. However, whether the reduction in fitness can be offset by the ability to withstand MBC need to be measured. 

The mechanism of resistance to carbendazim was often associated with point mutations in the *Tub* genes that change the structure of the fungicide binding site to, in turn, decrease sensitivity [18,38]. Analysis of the *U. virens* genome revealed two β-tubulin genes, *β1Tub* and *β2Tub*. Both of them were identified and investigated in this study. Most of the ascomycetes contain only one β-tubulin gene [39], while two genes have been found in some fungi, including *Aspergillus nidulans*, *Trichoderma* spp., *Colletotrichum* spp., and *Fusarium* spp. [40,41,42,43]. In our phylogenetic analyses, the ascomycetes that have two β-tubulin genes were also partially listed. 

The two tubulin genes in *U. virens* exhibited high sequence similarity. The deduced amino acid sequences of Uvβ1Tub and Uvβ2Tub were 78.12%. A similar high identity was also reported in *G. zeae*, with the identity of 76% [44]. However, in *U. virens*, whether their products have different functions need to be further investigated. Gene replacement demonstrated that the two genes are functionally interchangeable in *A. nidulans* [45] and *C. graminicola* [46]. In *G. zeae*, both tubulin isotypes function well by being assembled into cytoplasmic microtubules. In addition, the effects of β1Tub on mycelial growth, conidial germination, and pathogenicity have been verified by gene knockout in *G. zeae* [44]. MBC fungicides can interact with several regions of the tubulin molecule, including amino acid residues 6, 165 to 167, 198 to 200, 240, and 241 [44,47]. All these regions were conserved in Uvβ1Tub and Uvβ2Tub from sensitive isolates. 

In our study, as for the carbendazim-resistant isolates, point mutations were found in the *Uvβ1Tub* gene, not in the *Uvβ2Tub* gene. As for MBC resistance, substitutions at the *β2Tub* gene have frequently been reported to cause resistance in the field or laboratory isolates of several pathogenic fungi, including mutations at codons 6, 50, 134, 165, 167, 198, 200, 235, 240, 241, and 257 [15,18,21,23,48]. Variations in *β1Tub* have also been reported to be involved in the MBC resistance. In the case of UV-mutants of *F. moniliforme* (synonym, *F. verticillioides*), resistance resulted from a Tyr50Asp mutation in *β1Tub* [22]. The point mutations V91G and A411D found in *Uvβ1Tub* in our study have never before been reported in other pathogens. 

Other than point mutation, the majority of variations found in our study were non-sense mutations (isolates GL12b and ZJ7), codon mutations (isolates JR11 and YD8), and frameshift mutations (isolates GL23, HA17, XH5a, XH43b, and YZ11), which may cause the premature termination of *Uvβ1Tub* gene translation, opposing the formation of full-length transcripts, thereby impacting the gene expression. The increased resistance to benzimidazole fungicides in the *β1Tub* unexpressed mutants suggests *β1Tub* rather than *β2Tub* is the preferred binding target for MBC fungicides in *U. virens*. The binding preference of MBC with *β1Tub* rather than *β2Tub* has been verified in *F. graminearum* [43]. By gene knockout, MBC sensitivity was found to be significantly reduced in a *β1Tub* deletion isolate but increased in a *β2Tub* deletion isolate compared with that of a parental isolate, suggesting that *β1Tub* was involved in the MBC sensitivity of *F. graminearum* [43]. Homology modeling further verifies the possible MBC binding sites to be β1Tub rather than β2Tub. In addition, MBC fungicides are more likely to disrupt β1Tub microtubules rather than β2Tub microtubules in GFP–tub fusion mutants in vivo [43]. Thus, we assumed that the resistant mechanisms in those *U. virense* isolates seemed to be different from other pathogens. Unexpressed *β1Tub* gene may result in the lack of MBC target in *U. virens*, hence causing the resistance to the MBCs. For *U. virens*, the function and their involvement in MBC resistance of Uvβ1Tub and Uvβ2Tub need to be further studied. In addition, the reason why Uv*β1Tub* is so prone to mutate remains to be studied. 

We also confirmed the absence of mutations in the *β1Tub* and *β2Tub* sequence in the resistant isolates HA26, XH7b, and ZJ24, suggesting that there may be other pathways involved in the molecular mechanism of MBC resistance.

In conclusion, our study has revealed the emergence of the resistance of *U. virens* to MBC fungicides. We suggest that it is necessary to persistently monitor the resistance to MBCs. MBCs should be valuable when used in combination or alternation with other fungicides for the control of RFS in rice fields. 

## Figures and Tables

**Figure 1 jof-08-01311-f001:**
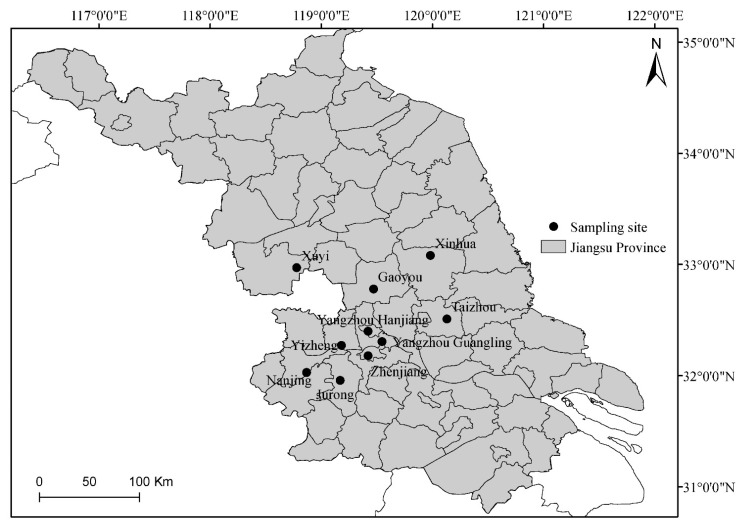
Geographic origins of *Ustilaginoidea virens* isolates collected in Jiangsu Province of East China.

**Figure 2 jof-08-01311-f002:**
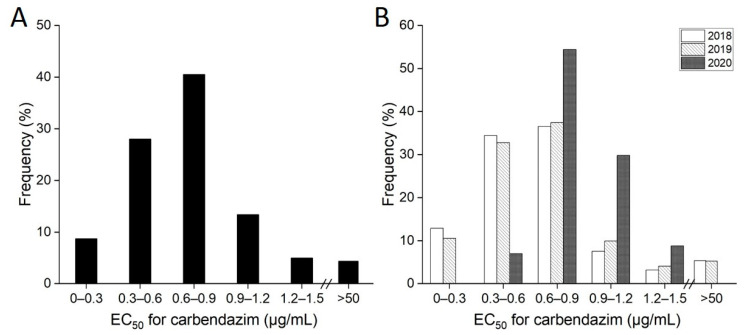
Frequency distribution of EC_50_ values of *Ustilaginoidea virens* isolates to carbendazim. (**A**) Frequency distribution of EC_50_ values of the total of 321 isolates. (**B**) Frequency distribution of EC_50_ values of isolates collected from different years.

**Figure 3 jof-08-01311-f003:**
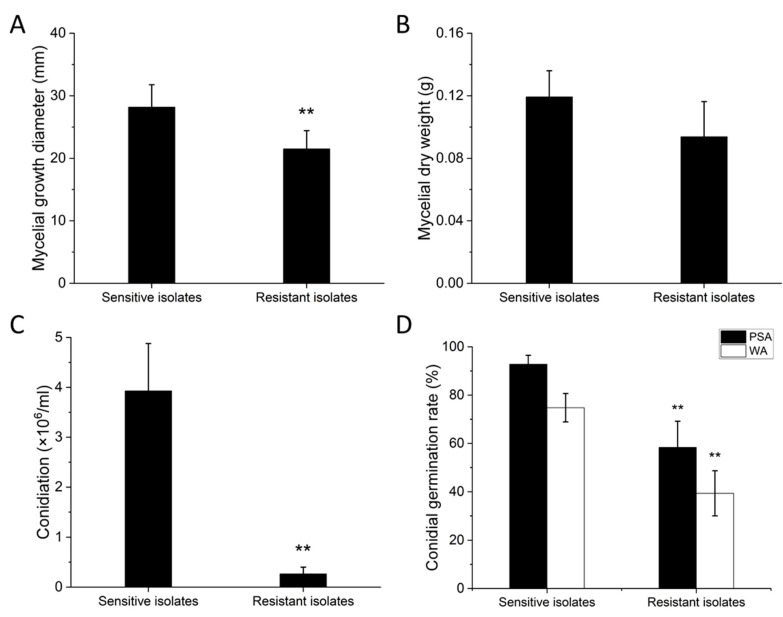
Biological characterization of carbendazim-sensitive and carbendazim-resistant *Ustilaginoidea virens* isolates. (**A**) Mycelial growth diameter, (**B**) mycelial dry weight, (**C**) conidiation, and (**D**) conidial germination rate. Five isolates (HWD, JS60-2, JY7b, JY11a, and JY30b) were tested as sensitive isolates, while 14 isolates (GL11, GL12b, GL23, HA17, HA26, JR11, JR12, XH5a, XH7b, XH43b, YD8, YZ11, ZJ7, and ZJ24) were tested as resistant isolates for mycelial growth diameter, mycelial dry weight, and conidiation. Five resistant isolates (GL11, HA17, XH5a, XH7b, and XH43b) were tested for conidial germination rate on PSA and WA media. Asterisks indicate that the difference is statistically significant. Error bars represent standard deviations.

**Figure 4 jof-08-01311-f004:**
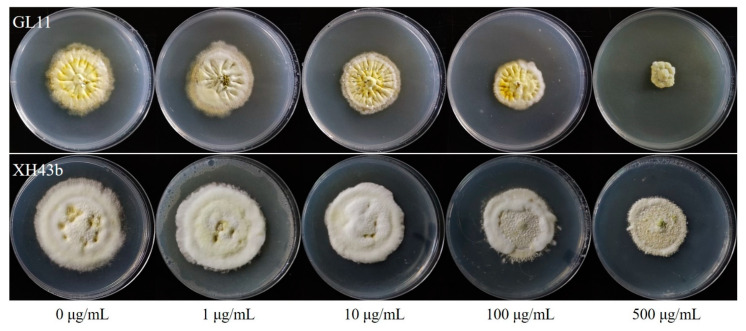
Sensitivity of resistant isolates GL11 and XH43b to carbendazim after 5 transfers (T5) on fungicide-free PSA. Photos were taken after 28 days of incubation at 27 °C in the dark.

**Figure 5 jof-08-01311-f005:**
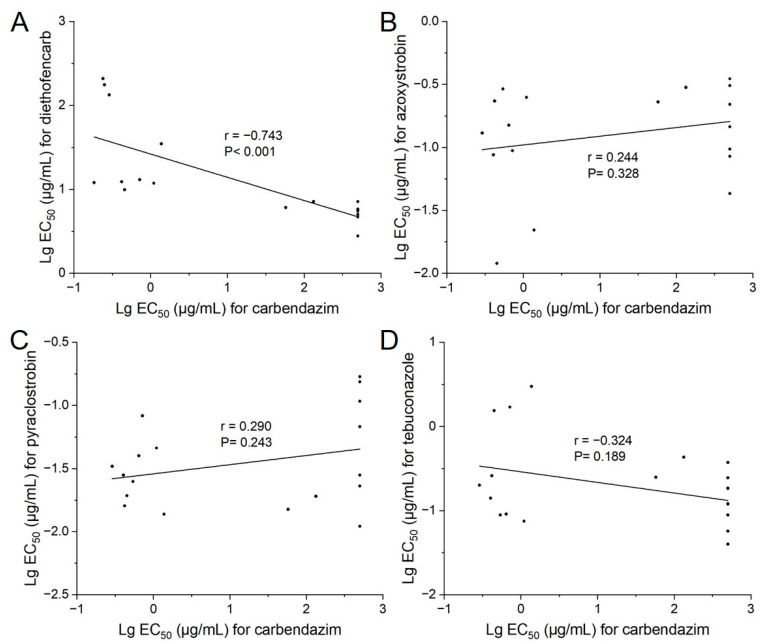
Correlation of log10-transformed concentration at which mycelial growth is inhibited 50% (EC_50_) values of *Ustilaginoidea virens* isolates for carbendazim and (**A**) diethofencarb, (**B**) azoxystrobin, (**C**) pyraclostrobin, and (**D**) tebuconazole.

**Figure 6 jof-08-01311-f006:**
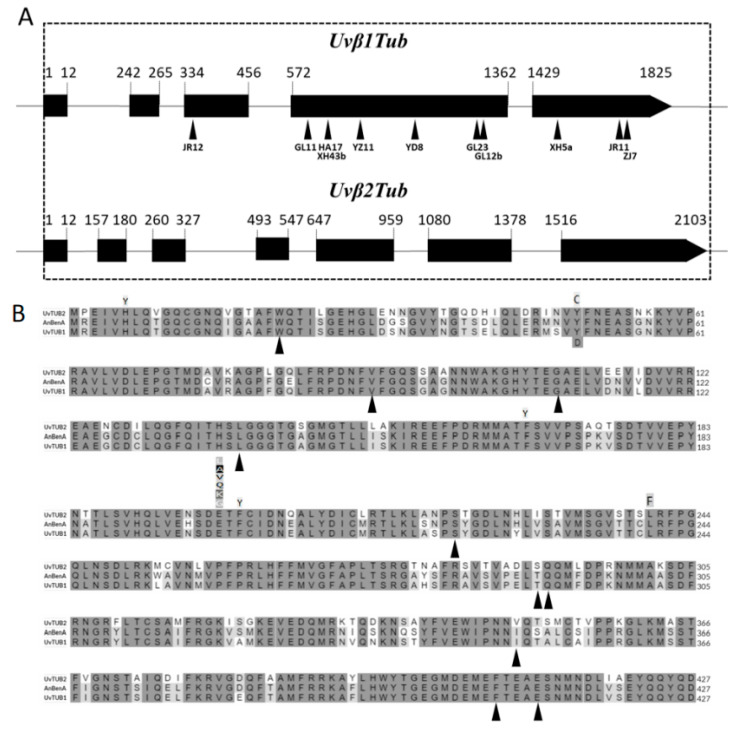
Schematic gene structure and alignments of *Ustilaginoidea virens Uvβ1Tub* and *Uvβ2Tub*. (**A**) *Uvβ1Tub* from twenty-three carbendazim-sensitive isolates is 1825 bp in size interrupted by four introns (with exons from 1–12, 242–265, 334–456, 572–1362, and 1429–1825). Locations of *Uvβ1Tub* variations in carbendazim-resistant isolates are marked with arrows. *Uvβ2Tub* from carbendazim-sensitive isolates is 2103 bp in size interrupted by six introns (with exons from 1–12, 157–180, 260–327, 493–547, 647–959, 1080–1378, and 1516–2103). Number indicates nucleotide position of the gene. (**B**) Alignments of β-tubulin amino acid sequences from *Ustilaginoidea virens and Aspergillus nidulans.* The sequences of deduced amino acids of β-tubulin from *A. nidulans* from the NCBI GenBank database AAA3328.1. The shaded letters indicate the conserved residues. Letters above the sequences: substitution amino acids in *β2Tub* from other resistant field isolates [21]. Letters under sequences: substitution amino acids in *β1Tub* from resistant laboratory mutants [22,23]. The codon positions where mutations occurred in *Uvβ1Tub* are marked with arrows.

**Figure 7 jof-08-01311-f007:**
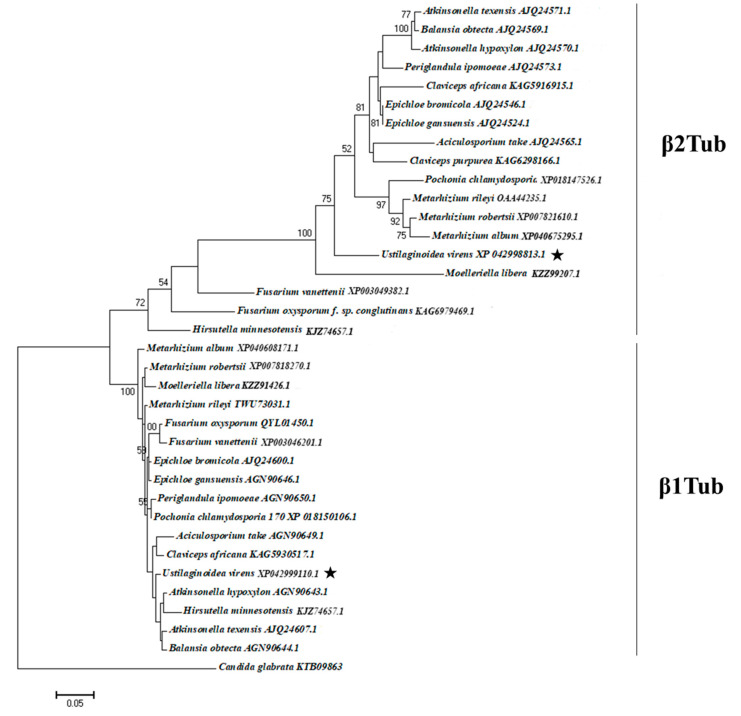
Phylogenetic tree generated by the maximum likelihood method with Mega 7.0 software [24] on the basis of deduced amino acid sequences of β-tubulin proteins. The deduced amino acid sequences of Uvβ1Tub and Uvβ2Tub for *Ustilaginoidea virens* isolate Uv8b and those from other fungal species, with 434 positions in the final dataset. Numbers labeled at each node indicate bootstrap values (%) from 1000 replicates. Stars indicate β1Tub and β2Tub of *Ustilaginoidea virens* in this study.

**Table 1 jof-08-01311-t001:** Carbendazim resistance in field isolates of *Ustilaginoidea virens* from different regions of Jiangsu Province.

Source ^x^	Number of Isolates	Resistant Isolates			Sensitive Isolates	
		Number (%) ^y^	EC_50_ (μg/mL)	Resistance Factor ^z^	EC_50_ (μg/mL)	Mean ± SEM
Gaoyou	53	0	-	-	0.143–1.269	0.671 ± 0.065
Jurong	28	2 (7.14%)	>500	>814	0.208–1.231	0.668 ± 0.061
Nanjing	10	0	-	-	0.291–1.361	0.805 ± 0.169
Taizhou	46	0	-	-	0.108–1.320	0.607 ± 0.081
Xinhua	53	3 (5.66%)	>500	>814	0.317–1.217	0.637 ± 0.042
Xuyi	35	2 (5.71%)	>500	>814	0.128–1.324	0.657 ± 0.076
Yizheng	16	1 (6.25%)	>500	>814	0.299–0.786	0.587 ± 0.077
YZ Guanglin	30	3 (10.00%)	57.6~ > 500	94~>814	0.155–1.378	0.774 ± 0.122
YZ Hanjiang	21	1 (4.76%)	>500	>814	0.278–0.983	0.703 ± 0.061
Zhenjiang	29	2 (6.90%)	>500	>814	0.355–1.360	0.637 ± 0.070
Total	321	14 (4.36%)	57.6~ > 500	94~>814	0.108–1.378	0.663 ± 0.031

^x^ YZ, Yangzhou. ^y^ Number and frequency (%) of resistant isolates. ^z^ Resistance factor = EC_50_ value of resistant isolate/The mean EC_50_ value of sensitive isolates.

**Table 2 jof-08-01311-t002:** Stability of resistant *Ustilaginoidea virens* isolates to carbendazim.

Isolate	Phenotype ^x^	EC_50_ (μg/mL) ^y^		MIC(μg/mL) ^z^	
		T_0_	T_5_	T_0_	T_5_
GL11	Carbendazim-R	57.6	68.1	>500	>500
GL12b	Carbendazim-R	133.3	114.9	>500	>500
Other isolates ^w^	Carbendazim-R	>500	>500	>500	>500

^w^ Other isolates: includes other carbendazim-resistant isolates GL23, HA17, HA26, JR11, JR12, XH5a, XH7b, XH43b, YD8, YZ11, ZJ7, and ZJ24. ^x^ R = Resistant. ^y^ The EC_50_ for the mycelial of initial generation (T_0_) and the last generation after 5 transfers on fungicide-free PSA (T_5_). ^z^ MIC = minimum inhibitory concentration.

**Table 3 jof-08-01311-t003:** Mutations of *Uvβ1Tub* in carbendazim resistant isolates of *Ustilaginoidea virens.*

Type of Mutation	Isolate	Mechanism	Change in *Uvβ1Tub*
Point mutation	GL11	V91G	T to G
	ZJ7	A411D	C to A
Non-sense mutation	GL12b	Q291 to stop	C to T
	JR12	W21 to stop	G to A
Codon mutation	JR11	codon 408–412 deletion	15 bp deletion
	YD8	S221 deletion	3 bp deletion
Frameshift mutation	GL23	frameshift	7 bp deletion
	HA17	frameshift	2 bp deletion
	XH5a	frameshift	1 bp deletion
	XH43b	frameshift	2 bp deletion
	YZ11	frameshift	2 bp deletion

## Data Availability

The data presented in this study are included in the article, further inquiries can be directed to the corresponding author.

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
