# Peer review of "Prevalence of Carbendazin Resistance in Field Populations of the Rice False Smut Pathogen Ustilaginoidea virens from Jiangsu, China, Molecular Mechanisms, and Fitness Stability"

_jof, 2022, doi:10.3390/jof8121311_

Round 1
Reviewer 1 Report
The study objectives were: (a) [to] determine the [levels and prevalence of] fungicide sensitivity [to the MBC fungicide carbendazim, mainly] of the rice false smut ascomycetous pathogen Ustilaginoidea virens from different fields in the major rice-producing areas [of the] Jiangsu province in China; (b) [to] characterize the carbendazim-resistant isolates; (c) to understand the molecular mechanisms [associated mainly with] carbendazim resistance.
The authors present the importance of the study very clearly. The general presentation of the manuscript, the significance of the study, the appropriate methods for hypothesis testing, the quality of data obtained, the summarization of the information, the figures presentation, and the very focused discussion pointed towards a recommendation for the acceptance of the manuscript after approaching few major concerns and implementing a minor revision, though.
1. The first major concern is that the study is focused on sensitivity / resistance to the MBC - carbendazim sensitivity even though assays and results from a series of sensitivity to several other fungicide groups such as QoIs and DMIs were included. See for example the paragraph from Material and Methods, that begins at line 109 and continues onwards: "Technical-grade carbendazim [...], azoxystrobin [...], pyraclostrobin [...], tebuconazole [...], and diethofencarb [...] were used in this study. This does not make sense unless the authors change the title and adjust the Materials and Methods properly by spliting the topic on sensitivity assay (perhaps creating a new topic) as to indicate that this was conducted with the aim of revealing any correlation / cross resistance among distinct fungicide groups. In addition, to indicate the rationale of this hypothesis testing, since very distinct fungicide groups were screened, none of which with reports of cross resistance with MBC. An important question is raised at this point: Wasn´t the true objective of this assay to search for evidence of multiple fungicide resistance in the pathogen´s populations instead of cross resistance?
2. The second major concern is related to the in vitro sensitivity testing assays. The authors reported on line 119: "Sensitivity to carbendazim, azoxystrobin, pyraclostrobin, tebuconazole and diethofencarb was assessed on fungicide-amended PSA medium at 0, 0.03, 0.1, 0.3, 1, 3, 10, 30, and 60 µg 120 a.i./mL." Please indicate if salicylhydroxamic acid (SHAM) was incorporated in the in vitro assays, which is required as an inhibitor of fungal alternative oxidase in the presence of the QoIs azoxystrobin and pyraclostrobin to avoid the inference of false positives for resistance to these fungicides. If this was not done in the QoI sensitivity assays, the results are flawed.
3. The third major concern is related to the results presented as topic 3.3. Cross-resistance analysis (line 254 onwards): Wasn´t the true objective of this assay to search for evidence of multiple fungicide resistance in the pathogen´s populations instead of cross resistance? Please clarify that.
4. The fourth major concerns the discussion of the resistance mechanisms associated with the MBC fungicide carbendazim in light of the current literature. I recommend the authors to implement a thorough comparison between mechanisms found for Ustilaginoidea virens in this study and the mechanisms described elsewhere for Aspergillus nidulans in a proposal for a unified nomenclature for target-site mutations associated with resistance to fungicides, including MBCs (published by Mair et al. 2016, available at https://pubmed.ncbi.nlm.nih.gov/27148866/)
5. My fifth major concern is about the complete lack of information [in the introduction] in regard to the harm of the fungicide carbendazim, which has been banned from several countries. The following paragraph has been extracted from a PAN Pesticide Action Network factsheet, from 2014, in regard the harm of carbendazim: "Independent literature shows that the pesticide carbendazim is a very dangerous "fungicide", capable of causing malformations in the fetus at very low doses and it's still uncertain if a safe level exists at all. Carbendazim is also capable of disrupting chromosome unfolding, can cause infertility of men and cancer". A review published in 2017 investigated the international regulatory situation of several pesticides, carbendazim included, in the following member countries: Organization for Economic Co-operation and Development (OECD), European Community, and the BRICS (Brazil, Russia, India, China, and South Africa). The fungicide carbendazim, in particular, was banned (specially for agricultural usage) in Japan, European Community, Iceland, Norway, Turkey and Switzerland, due to the fact that it is toxic to aquatic life (acutely and chronically); it is mutagenic (category 1B, i.e., a substance known to induce heritable genetic mutations in the germ cells of humans.) [according to the main damages classification criteria – European Directive n. 1,272/2008], and it affects the health and the environment causing endocrine disruption in wildlife and cancer in humans [check reference https://doi.org/10.1590/0102-311X00061820 ]. Carbendazim has been banned also in the USA since 2012 and in the UK since 2017. Despite the harm of carbendazim, even nowadays, products banned in other countries are still used in developing countries such as Brazil, China and India. Brazil banned carbendazim just recently, in 2022. Therefore, it is strongly recommended that the authors approach, in the introduction, this reasonable concern about the harm of the current use of the fungicide carbendazim on agricultural crops such as rice, since it has been banned worldwide and eventually will be banned in China too. And despite all these concerns, bring also relevant information on why this fungicide is still used and the reason why the harm of carbendazim was ignored.
6. And finally, it concerned me that the authors did not approach any anti-resistance strategy (either anti-emergence or anti-resistance per se) that should / or have been implemented in the field, since MBC fungicides are considered high risk fungicides for the emergence of resistance in plant pathogen´s population worldwide when deployed singly in agricultural fields (check Hobelen et al . 2014, available at https://pubmed.ncbi.nlm.nih.gov/24658678/, as a reference to support this relevant discussion).
The minor revision includes:
Describe in detail which locations are counties and which locations are cities from the Jiangsu province as most of the readers might not be familiar with China's geography and won't be able to discriminate amongst the locations. Therefore, please, fix the following paragraph at line 89 onwards and on Table 1, to clarify the geographical locations, and spell out the meaning of YZ: "The rice field are located in 10 counties and cities, namely Gaoyou, Jurong, Nanjing, Taizhou, Xinhua, Xuyi, Yizheng, Yangzhou Guanglin (YZ Guanglin), Yangzhou Hanjiang (YZ Hanjiang) and Zhenjiang''. A detailed map would certainly bring more clarity to the geographical sampling of the pathogen´s populations.
Author Response
Comments and Suggestions for Authors
The study objectives were: (a) [to] determine the [levels and prevalence of] fungicide sensitivity [to the MBC fungicide carbendazim, mainly] of the rice false smut ascomycetous pathogen Ustilaginoidea virens from different fields in the major rice-producing areas [of the] Jiangsu province in China; (b) [to] characterize the carbendazim-resistant isolates; (c) to understand the molecular mechanisms [associated mainly with] carbendazim resistance.
The authors present the importance of the study very clearly. The general presentation of the manuscript, the significance of the study, the appropriate methods for hypothesis testing, the quality of data obtained, the summarization of the information, the figures presentation, and the very focused discussion pointed towards a recommendation for the acceptance of the manuscript after approaching few major concerns and implementing a minor revision, though.
Response: Thank you very much for your comments and suggestions to improve the manuscript.
- The first major concern is that the study is focused on sensitivity / resistance to the MBC - carbendazim sensitivity even though assays and results from a series of sensitivity to several other fungicide groups such as QoIs and DMIs were included. See for example the paragraph from Material and Methods, that begins at line 109 and continues onwards: "Technical-grade carbendazim [...], azoxystrobin [...], pyraclostrobin [...], tebuconazole [...], and diethofencarb [...] were used in this study. This does not make sense unless the authors change the title and adjust the Materials and Methods properly by spliting the topic on sensitivity assay (perhaps creating a new topic) as to indicate that this was conducted with the aim of revealing any correlation / cross resistance among distinct fungicide groups. In addition, to indicate the rationale of this hypothesis testing, since very distinct fungicide groups were screened, none of which with reports of cross resistance with MBC. An important question is raised at this point: Wasn´t the true objective of this assay to search for evidence of multiple fungicide resistance in the pathogen´s populations instead of cross resistance?
Response: According to your suggestion, we adjusted the Materials and Methods 2.3. to “2.3. In vitro sensitivity determination of U. virens to carbendazim”, and “2.6. Cross-resistance analysis” to “2.6. Correlation analysis in sensitivity of carbendazim with diethofencarb, azoxystrobin, pyraclostrobin and tebuconazole”, and added relevant contents.
- The second major concern is related to the in vitro sensitivity testing assays. The authors reported on line 119: "Sensitivity to carbendazim, azoxystrobin, pyraclostrobin, tebuconazole and diethofencarb was assessed on fungicide-amended PSA medium at 0, 0.03, 0.1, 0.3, 1, 3, 10, 30, and 60 µg 120 a.i./mL." Please indicate if salicylhydroxamic acid (SHAM) was incorporated in the in vitro assays, which is required as an inhibitor of fungal alternative oxidase in the presence of the QoIs azoxystrobin and pyraclostrobin to avoid the inference of false positives for resistance to these fungicides. If this was not done in the QoI sensitivity assays, the results are flawed.
Response: In our previous studies, all tested U. virens isolates were sensitive to two QoI fungicides azoxystrobin and pyraclostrobin without SHAM (https://doi.org/10.1094/PDIS-12-21-2850-RE), and SHAM at lower concentration showed strong toxicity to U. virens, inhibiting mycelial growth, conidial germination, peroxidase (POD) and esterase activity of U. virens (https://doi.org/10.3390/jof8111231).
We added it as “In sensitivity determination to azoxystrobin and pyraclostrobin, salicylhydroxamic acid (SHAM) was not included in medium, since all tested isolates were sensitive to these two QoI fungicides and SHAM showed strong toxicity to U. virens according to our previous studies.”
- The third major concern is related to the results presented as topic 3.3. Cross-resistance analysis (line 254 onwards): Wasn´t the true objective of this assay to search for evidence of multiple fungicide resistance in the pathogen´s populations instead of cross resistance? Please clarify that.
Response: Yes, the objective of this assay was to search for evidence of multiple fungicide resistance in the pathogen´s populations. So we changed “3.3. Cross-resistance analysis” to “3.3. Correlation in sensitivity of carbendazim with diethofencarb, azoxystrobin, pyraclostrobin and tebuconazole”.
- The fourth major concerns the discussion of the resistance mechanisms associated with the MBC fungicide carbendazim in light of the current literature. I recommend the authors to implement a thorough comparison between mechanisms found for Ustilaginoidea virens in this study and the mechanisms described elsewhere for Aspergillus nidulans in a proposal for a unified nomenclature for target-site mutations associated with resistance to fungicides, including MBCs (published by Mair et al. 2016, available at https://pubmed.ncbi.nlm.nih.gov/27148866/)
Response: We have implemented a thorough comparison between Ustilaginoidea virens and Aspergillus nidulans according to the reference (https://pubmed.ncbi.nlm.nih.gov/27148866/). Our result is consistent with a unified nomenclature for target-site mutations associated with resistance to MBC fungicides. The following sentence has been extracted from the reference: “The alignments for b-tubulin and Cytb are essentially collinear in fungi studied to date, and hence there are no changes to be made to the current nomenclature.”
- My fifth major concern is about the complete lack of information [in the introduction] in regard to the harm of the fungicide carbendazim, which has been banned from several countries. The following paragraph has been extracted from a PAN Pesticide Action Network factsheet, from 2014, in regard the harm of carbendazim: "Independent literature shows that the pesticide carbendazim is a very dangerous "fungicide", capable of causing malformations in the fetus at very low doses and it's still uncertain if a safe level exists at all. Carbendazim is also capable of disrupting chromosome unfolding, can cause infertility of men and cancer". A review published in 2017 investigated the international regulatory situation of several pesticides, carbendazim included, in the following member countries: Organization for Economic Co-operation and Development (OECD), European Community, and the BRICS (Brazil, Russia, India, China, and South Africa). The fungicide carbendazim, in particular, was banned (specially for agricultural usage) in Japan, European Community, Iceland, Norway, Turkey and Switzerland, due to the fact that it is toxic to aquatic life (acutely and chronically); it is mutagenic (category 1B, i.e., a substance known to induce heritable genetic mutations in the germ cells of humans.) [according to the main damages classification criteria – European Directive n. 1,272/2008], and it affects the health and the environment causing endocrine disruption in wildlife and cancer in humans [check reference https://doi.org/10.1590/0102-311X00061820 ]. Carbendazim has been banned also in the USA since 2012 and in the UK since 2017. Despite the harm of carbendazim, even nowadays, products banned in other countries are still used in developing countries such as Brazil, China and India. Brazil banned carbendazim just recently, in 2022. Therefore, it is strongly recommended that the authors approach, in the introduction, this reasonable concern about the harm of the current use of the fungicide carbendazim on agricultural crops such as rice, since it has been banned worldwide and eventually will be banned in China too. And despite all these concerns, bring also relevant information on why this fungicide is still used and the reason why the harm of carbendazim was ignored.
Response: We added some information in regard to the harm of the fungicide carbendazim in the introduction as “In addition, carbendazim may cause endocrine disruption in wildlife and cancer in humans. Carbendazim has been banned in some countries such as USA and UK. Therefore, the application of carbendazim in agriculture might be restricted or prohibited in China in the future”.
- And finally, it concerned me that the authors did not approach any anti-resistance strategy (either anti-emergence or anti-resistance per se) that should / or have been implemented in the field, since MBC fungicides are considered high risk fungicides for the emergence of resistance in plant pathogen´s population worldwide when deployed singly in agricultural fields (check Hobelen et al . 2014, available at https://pubmed.ncbi.nlm.nih.gov/24658678/, as a reference to support this relevant discussion).
Response: We proposed some resistance management strategies in the third paragraph of the Discussion. “The correlation analysis assay showed that the 9 resistant isolates were sensitive to azoxystrobin, pyraclostrobin, tebuconazole, indicating that there was no multiple fungicide resistance between MBC, QoI, and DMI fungicides. Thus, QoI and DMI fungicides can be used for the management of carbendazim resistance in U. virens. Moreover, the mixture of carbendazim with DMIs, such as tebuconazole (reg-istration number: PD20110332), tradimefon (registration number: PD20060057), hexaconazoel (registration number: PD20181518), which has been registered to control rice diseases in China could still be used to manage the resistance of U. virens to MBC at present.”
The minor revision includes:
Describe in detail which locations are counties and which locations are cities from the Jiangsu province as most of the readers might not be familiar with China's geography and won't be able to discriminate amongst the locations. Therefore, please, fix the following paragraph at line 89 onwards and on Table 1, to clarify the geographical locations, and spell out the meaning of YZ: "The rice field are located in 10 counties and cities, namely Gaoyou, Jurong, Nanjing, Taizhou, Xinhua, Xuyi, Yizheng, Yangzhou Guanglin (YZ Guanglin), Yangzhou Hanjiang (YZ Hanjiang) and Zhenjiang''. A detailed map would certainly bring more clarity to the geographical sampling of the pathogen´s populations.
Response: A detailed map was added as Figure 1.
Reviewer 2 Report
Dear Authors,
The manuscript is interesting, the data analysis are in accordance with the conclusions and the writing is easy to follow. I recommend the acceptance of the manuscript .
Author Response
Comments and Suggestions for Authors
Dear Authors,
The manuscript is interesting, the data analysis are in accordance with the conclusions and the writing is easy to follow. I recommend the acceptance of the manuscript .
Response: Thank you very much for your kindly comments and suggestions.
Reviewer 3 Report
Till now management of false smut disease of rice is mostly dependent on the prophylactic spray of fungicides. Carbendazim is one of the best broad-spectrum widely available fungicides. So, this article is quite informative and has given a signal on the application of carbendazim to manage FS disease. But it is not understandable how resistant isolates give less growth, conidiation, and germination ability than sensitive isolates.
Author Response
Comments and Suggestions for Authors
Till now management of false smut disease of rice is mostly dependent on the prophylactic spray of fungicides. Carbendazim is one of the best broad-spectrum widely available fungicides. So, this article is quite informative and has given a signal on the application of carbendazim to manage FS disease. But it is not understandable how resistant isolates give less growth, conidiation, and germination ability than sensitive isolates.
Response: Thank you very much for your kindly comments and suggestions. We mentioned relevant information in the Discussion. Firstly, similar to our results, fitness reduction was also observed in F. fujikuroi MBC resistant isolates in terms of fewer conidia and less virulence. Secondly, variations including point mutations, non-sense mutations, codon mutations, and frameshift mutations were found in the Uvβ1Tub gene from the 14 carbendazim-resistant isolates. The variations impair the normal function of the Uvβ1Tub gene. In G. zeae, both tubulin isotypes function well by being assembled into cytoplasmic microtubules. The effects of β1Tub on mycelial growth, conidial germination, and pathogenicity have been verified by gene knockout in G. zeae. Finally, in an experiment we are conducting, knockout of Uvβ1Tub gene results in resistance to carbendazim with reduction in fitness.
Reviewer 4 Report
Dear Colleagues. There are several questions and wishes. They are in the attached file.

Author Response
Line 88
“Isolates of U. virens used in this study were obtained during 2018 to 2020 from Jiangsu”
It is worth mentioning that resistance was compared across different years. Moreover, there are data on line 220, Figure 1. What can the data on resistance to carbedazim in isolates of different years mean?
Response: Thank you very much for your kindly comments and suggestions. As shown in the Results 3.1., the data means “The frequency distribution of the EC50 values for carbendazim was unimodal (Figure 2A). In terms of years, although no resistant isolates were detected in 2020, the proportion of isolates insensitive to carbendazim in 2020 was higher than that in 2018 and 2019 (Figure 2B).”
Line 155-159
“Stability of resistant to carbendazim Mycelium plugs were taken from the periphery of the colonies and transferred to fresh fungicide-free PSA, then incubated the plate at 27℃ for 10 days. The process was repeated for 5 generations but new plates were always inoculated with the colony of the previous generation. EC50 values for carbendazim were established before the first and after the last transfer, as described above”.
It would be good to illustrate this material with Figures or photographs of the colonies. Was it only 5 passages? Did you try to explore a longer period? How is the resistance in the field? Are there regional differences?
Response: Thanks for your good suggestion. We added Figure 4 to illustrate the stability of resistant to carbendazim. According to our previous study, the false resistance to fungicide pyraclostrobin disappeared after 5 generations (https://doi.org/10.1094/PDIS-12-21-2850-RE). Therefore, we transferred 5 generations in this study. In fact, in order to maintain the activity of the isolates, we need to subculture each isolate within a month. Up to now, the resistant isolates have been subcultured for more than 30 generations in the laboratory, and the resistance to carbendazim is still stable. Since most of resistant isolates produced no or little conidia, they lost pathogenicity by artificial inoculation, which could not to determine the resistance in the field. Among the resistant isolates, there was no significant difference in mean EC50 values and fitness of isolates from different regions.
Line 225
«Four fitness components were tested for all 14 resistant isolates and five sensitive isolates. Overall, fitness penalties were recorded for the resistant isolates. The mycelial growth diameters of the resistant isolates were significantly lower than the sensitive isolates (Figure 2A). Also, mycelail – misprint dry weight was lower in the resistant isolates, but difference was statistically non-significant…».
Response: Sorry for that mistake, we revised it.
Figure 2. Biological characterization of carbendazim-sensitive and carbendazim-resistant Ustilagnoidea virens isolate.
Are there similar data on the medium with fungicide? How does the fungicide work on resistant isolates?
Response: No, the fitness of sensitive isolates will be significantly inhibited on the medium with fungicide. As shown in Figure 4, carbendazim at low concentration did not affect the fitness of resistant isolates.
Line 119-121
“Sensitivity to carbendazim, azoxystrobin, pyraclostrobin, tebuconazole and dietho fencarb was assessed on fungicide-amended PSA at 0, 0.03, 0.1, 0.3, 1, 3, 10, 30, and 60 µg 120 a.i./mL.”
Line 262
In my opinion, the data on the correlation of resistance with other fungicides, presented in the additional Figure 1, are of interest. They should be included in the main text, next to Figure 3.
Response: We adjusted this Figure based on your suggestion.
Round 2
Reviewer 1 Report
I considered that the major suggestions of changes from the first round of the review were mostly attended by the authors. However, before recommending acceptance I suggest a slight change in the title to reflect more appropriately the outcome of their research:
Prevalence of carbendazin resistance in field populations of the rice false smut pathogen Ustilaginoidea virens from Jiangsu, China, molecular mechanisms and fitness stability.
Also, I considered that the manuscript still lacks the recommendation of two key anti-resistance strategies: a) to limit, in regional scale, carbendazin fungicide sprays to a minimum and only in mixtures or co-formulation with multi-site, broad spectrum, protectant, low-risk fungicides (such as mancozeb, chlorothalonil , e.g.), given the evidence that the resistant strains of the pathogen could be dispersed long distance from a source of resistant inoculum (line 383, onwards). b) to discourage individual decisions to use only the systemic, single site, high risk fungicides, such as MBCs and QoIs. You can use the reference to support this asertion (https://www.mdpi.com/2073-4395/12/12/2952; look at Discussion, 9th paragraph).
On line 385, fix the spelling of triadimefon and hexaconazole.
Author Response
I considered that the major suggestions of changes from the first round of the review were mostly attended by the authors. However, before recommending acceptance I suggest a slight change in the title to reflect more appropriately the outcome of their research:
Prevalence of carbendazin resistance in field populations of the rice false smut pathogen Ustilaginoidea virens from Jiangsu, China, molecular mechanisms and fitness stability.
Also, I considered that the manuscript still lacks the recommendation of two key anti-resistance strategies: a) to limit, in regional scale, carbendazin fungicide sprays to a minimum and only in mixtures or co-formulation with multi-site, broad spectrum, protectant, low-risk fungicides (such as mancozeb, chlorothalonil , e.g.), given the evidence that the resistant strains of the pathogen could be dispersed long distance from a source of resistant inoculum (line 383, onwards). b) to discourage individual decisions to use only the systemic, single site, high risk fungicides, such as MBCs and QoIs. You can use the reference to support this asertion (https://www.mdpi.com/2073-4395/12/12/2952; look at Discussion, 9th paragraph).
On line 385, fix the spelling of triadimefon and hexaconazole.
Response: Thank you very much for your professionalism and carefulness. According to your suggestions, we revised the title, fixed the spelling of triadimefon and hexaconazole, and added the anti-resistance strategies and the relevant reference in the Discussion.
Reviewer 3 Report
I have differences with the concept. As per my understanding and working experiences, resistant isolates (to fungicides) give higher mycelial growth, so the dry weight is more, more conidiation and germinability than the sensitive isolates but you explained completely opposite results. In the case of biotrophic fungi (eg. powdery mildew of grapes), the experiment is usually done in situ, and the result is somewhat different from culturable fungi. Can you explain why resistant isolates of U. virens give lower growth and condiation than sensitive isolates?